# Microbial Synthesis of High-Molecular-Weight, Highly Repetitive Protein Polymers

**DOI:** 10.3390/ijms24076416

**Published:** 2023-03-29

**Authors:** Juya Jeon, Shri Venkatesh Subramani, Kok Zhi Lee, Bojing Jiang, Fuzhong Zhang

**Affiliations:** 1Department of Energy, Environmental and Chemical Engineering, Washington University in St. Louis, Saint Louis, MO 63130, USA; jjuya@wustl.edu (J.J.); s.shrivenkatesh@wustl.edu (S.V.S.); kokzhi@wustl.edu (K.Z.L.); bojing.jiang@wustl.edu (B.J.); 2Institute of Materials Science and Engineering, Washington University in St. Louis, Saint Louis, MO 63130, USA; 3Division of Biological & Biomedical Sciences, Washington University in St. Louis, Saint Louis, MO 63130, USA

**Keywords:** repetitive proteins, microbial synthesis, silk, high molecular weight, protein polymers, synthetic biology

## Abstract

High molecular weight (MW), highly repetitive protein polymers are attractive candidates to replace petroleum-derived materials as these protein-based materials (PBMs) are renewable, biodegradable, and have outstanding mechanical properties. However, their high MW and highly repetitive sequence features make them difficult to synthesize in fast-growing microbial cells in sufficient amounts for real applications. To overcome this challenge, various methods were developed to synthesize repetitive PBMs. Here, we review recent strategies in the construction of repetitive genes, expression of repetitive proteins from circular mRNAs, and synthesis of repetitive proteins by ligation and protein polymerization. We discuss the advantages and limitations of each method and highlight future directions that will lead to scalable production of highly repetitive PBMs for a wide range of applications.

## 1. Introduction

Humans have thousands of years of history in using protein-based materials (PBMs), including wool, leather, and silk. These PBMs are renewable, biodegradable, and some display attractive mechanical and biological properties that are difficult to replicate even in modern synthetic polymeric materials [1,2,3]. PBMs with advantageous mechanical properties are particularly useful as they can potentially replace petroleum-derived fibers, films, and plastics, thus opening a wide range of applications. However, to meet the large demands of PBMs for modern applications, harvesting PBMs from natural sources is often no longer economical or is impractical due to the limited material amounts from natural production hosts, complicated purification procedures, and processing steps that may alter the PBMs’ mechanical properties [2,4,5].

For example, spider silk fibers display a unique combination of high tensile strength and toughness and were recognized decades ago as candidates to replace nylon and Kevlar for mechanically demanding applications [6]. Unfortunately, large-scale silk production by farming spiders is not feasible due to their territorial and cannibalistic behaviors. As a result, researchers have been actively searching for alternative bioproduction strategies for spider silk fiber. A wide range of bioproduction hosts, from bacteria to yeasts, insects, goats, and other mammalian cell lines, have been explored for silk protein production using recombinant silk DNAs [5]. Among these heterologous hosts, rapid-growing bacteria, such as *Escherichia coli*, are particularly attractive due to their abilities including metabolizing cheap feedstock, rapid biomass growth, well-known physiological and genomic information, and ease of genetic engineering [4,5]. More importantly, recent advances in protein engineering and synthetic biology have created artificial PBMs that display properties even beyond the best-performing natural PBMs [7,8,9]. For example, artificially designed silk-amyloid hybrid proteins that can be produced in engineered bacteria and spun to fibers with both strength and toughness higher than some natural spider silk fibers [9,10]. Therefore, there has been a huge need and interest in manufacturing PBMs from microbial hosts.

Similar to synthetic polymers, mechanically advantageous PBMs often have high molecular weight (MW) and their sequences consist of repeated amino acid sequences [11]. Such sequence features have brought major challenges in their biosynthesis, particularly in heterologous microbial hosts [12,13]. First, genes that encode high-MW, highly repetitive proteins are encoded by long and repetitive genes that are difficult to clone [14]. Second, these genes are often unstable in heterologous hosts and undergo recombination that permanently removes part of the repeating genes. The resulting gene fragments either do not translate or only translate truncated PBMs that do not have the desirable mechanical properties. Third, high-MW and highly repetitive proteins often have very low yield due to mRNA instability, translational jam, codon bias, translational burden, and other issues [15,16]. As a result, expressing high-MW, highly repetitive PBMs in large quantities and at high yields using traditional recombination technology is extremely challenging.

To overcome these challenges, multiple modern synthetic biology techniques have been developed, enabling the biosynthesis of sufficient proteins to support material research. This review focuses on recent advances in (1) construction of repeated gene sequences for repetitive proteins, (2) expression of repetitive proteins from circular mRNA, and (3) post-translational ligation of relatively small, genetically stable protein subunits into high-MW proteins.

## 2. Construction of Repetitive Genes for Repetitive Protein

Repeated proteins found in nature serve many functional roles to their hosts. To date, many repeated proteins, including silk, elastin, and squid protein, have been expressed in bacteria [7,12,13,17]. Due to limited codon choices, genes coding for repetitive proteins are unavoidably repetitive. Traditional polymerase chain reaction (PCR)-based cloning strategies do not work for repetitive DNAs because primers may randomly anneal to any repeat sequence. Restriction-enzyme based methods are also challenging, as restriction sites may exist in every single repeat. This section will discuss modern synthetic biology strategies to construct repetitive genes that encode repetitive PBMs. 

It should provide a concise and precise description of the experimental results, their interpretation, as well as the experimental conclusions that can be drawn.

### 2.1. Golden Gate DNA Assembly

Golden Gate DNA assembly uses type II-S restriction enzymes (e.g., BsaI), which recognize and cleave DNA at different sites, thus allowing the same repetitive DNA fragments to have different sticky ends [18] (Figure 1a). After type II-S enzyme digestion, multiple DNA fragments carrying unique overhang sequences can be ligated together with controlled order and orientation. Additionally, restriction enzyme recognition sites can be removed during digestion, making the ligation products not digestible by the same type II-S restriction enzyme. Thus, multiple DNA fragments can be precisely ligated by one-pot, multi-step, digestion–ligation reactions using Golden Gate assembly, yielding repetitive DNA products with defined DNA sequences.

Golden Gate assembly has been successfully used to construct genes for amyloid-spider silk proteins, elastin-like polypeptides, and squid ring-teeth proteins [7,13,17]. Using this method, up to ten highly repetitive DNA fragments can be ligated in one step. Although assembly of more fragments is still possible, increasing the number of DNA fragments decreases the success rate. Although powerful, Golden Gate DNA assembly is limited by the availability of unique overhang sites flanking each DNA fragment. Alternatively, multiple rounds of Golden Gate assembly can be used to construct genes with more than a dozen of repeats.

### 2.2. Rolling Circle Amplification

Rolling circle amplification relies on continuous elongation of circular single-stranded DNA that results in DNA products with multiple repeats of the amplified sequence [13] (Figure 1b). In this method, a single-stranded DNA that encodes one protein repeat will first be synthesized and circularized. The rolling circle amplification can be initiated by adding together a PCR reaction mixture and a primer that anneals the circular DNA. The amplified product will then be subjected to thermal denaturation and annealing followed by extension to obtain double-stranded DNAs. The products are a mixture of DNA fragments with varying number of repeats, usually ranging from 4 to 11 repeats depending on the size of the DNA monomer [13]. This DNA mixture can then be cloned to an expression vector for selecting genes with desirable sizes [13].

Because this method involves annealing between repeated sequences, mis-annealing frequently occurs and leads to frameshift on the coding sequence. Such a problem cannot be easily detected due to challenges in DNA sequencing. Additionally, the lack of sequencing makes it difficult to precisely determine the number of DNA repeats. Very often, the exact gene sizes can only be determined when their encoding proteins are expressed, purified, and subjected to mass spectrometry analysis, which is too late if the gene is wrong.

### 2.3. Combinatorial Codon Scrambling for DNA Synthesis and Amplification

To further assist expression of repetitive PBMs, a codon-scrambling algorithm was developed to reduce the repetitiveness of DNA sequences [19] (Figure 1c). Given a target protein sequence, the codon-scrambling algorithm searches for a different combination of synonymous codons for each repetitive protein fragment, with the goal to reduce repetitiveness at the DNA level. This method was successful in synthesizing a wide range of repeated protein polymers, including elastin-like polypeptides, resilin-like polypeptide, and collagen-like polypeptide, with the number of repeats up to 150 [19]. The features of these DNA assembly methods are summarized in Table 1.

Although these methods are successful in constructing genes for repetitive proteins, transforming these long and repetitive DNA sequences to heterologous production hosts still presents problems. Homologous recombination frequently occurs in many microbial hosts, leading to undesired truncations [20]. Furthermore, leaky expression or unwanted expression from hidden promoters within the long DNA sequence cause burden to host cells, which in turn increases the mutation rate and prevents PBM expression. Additionally, construction of high-MW PBMs with hundreds of repeats (e.g., spider silk) still requires multiple assembly steps or a combination of multiple strategies, making it laborious and time inefficient. Thus, more efficient methods have been developed to avoid the use of long and repetitive DNAs.

## 3. Expression of Repetitive Proteins from Circular mRNA

One smart strategy that avoids the use of long and repetitive DNA is repetitive protein translation from circular mRNA (cmRNA). cmRNAs are cyclized, single-stranded RNA derived from back-splicing of precursor mRNAs. A cmRNA can be generated by fusion of a self-splicing intron (e.g., the *td* gene from T4 phage) to a mRNA that encodes the target protein [21,22]. Once transcribed, the self-splicing intron catalyzes a spontaneous splicing–ligation reaction to form cmRNA that contains a stop-codon-free coding RNA (Figure 2). During translation, a ribosome translates through the cmRNA rounds and rounds until it disengages from the cmRNA template, producing repetitive proteins.

### 3.1. Advantageous in PBM Production Using cmRNAs

The use of cmRNA allows the synthesis of repetitive proteins from short, non-repetitive DNA and thus offers multiple advantages. First, it effectively avoids the need for constructing repetitive DNAs and issues associated with repetitive genes such as genetic instability. Second, because cmRNAs do not have free 5′ and 3′ ends, which are recognition sites of common ribonucleases, cmRNAs are more stable (less prone to RNase degradation) than their linear mRNA counterparts. The higher RNA stability allows more proteins to be translated per mRNA (Figure 2) [21,22].

### 3.2. Examples of PBM Production from cmRNAs

The advantages of cmRNAs have encouraged material scientists to explore them for synthesis of repetitive PBMs. Lee et al. first engineered cmRNAs to express spider dragline silk protein MaSp1 from *Trichonephila clavipes* in *E. coli* [21]. Although cmRNA encoding short 16× silk repeats (48.5 kDa) was successfully transcribed, the translated silk proteins have less than 32 repeats, meaning translation terminated before completing one around of translation and protein yield was very low [21]. Success was later obtained by Liu et al. who used cmRNA that contained a relatively shorter MaSp1sequence (encoding a 5.1 kDa protein monomer). In this work, the RBS together with the start codon were placed downstream of the silk coding sequence to prevent protein expression from un-circularized mRNA. The expressed protein products reached a maximal MW of 110 kDa, representing continuous translation from the cmRNA for at least 22 rounds. The protein yield reached 22.1 mg/L. The authors further used cmRNA to express a flagelliform silk and obtained similar success [20].

### 3.3. Challenges and Opportunities in Producing PBMs from cmRNAs

While promising, producing PBMs from cmRNA also has several disadvantages and limitations. Firstly, current RNA cyclization efficiency is low, particularly for mRNAs that form complex secondary structures [20]. RNA cyclization efficiency also depends on RNA length, with the most optimal length being between 300 and 500 base pairs, potentially due to higher entropy cost associated with cyclizing longer RNAs [23]. As a result, any protein repeat that has less than 100 or more than 170 residues may have low cyclization efficiency [20]. Second, translation termination is not controlled by stop codons but by random disengagement of the ribosome from cmRNA. Therefore, the resulting PBMs are a mixture of proteins with random C-termini without a defined MW, which may affect material properties. Lastly, translation efficiency can be strongly affected by unwanted interactions between target proteins. If the produced proteins strongly self-associate or aggregate, their interaction may make the ribosome disengage from the cmRNA template, resulting in early translation termination and low-MW products, which do have desirable material properties [21].

Although facing these challenges, cmRNA may still be valuable for the biosynthesis of repetitive PBMs upon further engineering. For example, to prevent early disengagement due to protein self-interaction, target proteins can be potentially engineered to reduce aggregation propensity, (e.g., substituting hydrophobic residues with hydrophilic residues). Furthermore, cmRNA may be found to be more suitable for expressing hydrophilic proteins that have relatively fewer numbers of repeats, such as fibronectins and collagens.

## 4. Synthesis of Repetitive Proteins Using Protein Ligation or Polymerization

High-MW, highly repetitive PBMs can be also synthesized by ligation or polymerization of low-MW, non-repetitive proteins (hereafter called protein monomers) through post-translational reactions. Expression of non-repetitive protein monomers avoids all the above-discussed problems (e.g., construction of repetitive DNAs, genetic instability, and RNA cyclization efficiency, etc.) and can be expressed at high levels. Here, we discuss these strategies by dividing them into two categories: (1) ligation of the protein backbones by forming new amide bonds and (2) crosslinking of protein monomers using side-chain chemistry. These strategies can either be used individually or in combination to create high-MW protein products.

### 4.1. Ligation of the Protein Backbones via Peptide Bonds

#### 4.1.1. Split Intein (SI)-Mediated Protein Ligation

Inteins are unique protein domains located in the middle of other protein sequences (called exteins). Inteins catalyze spontaneous splicing–ligation reactions that cleave them out and covalently ligate their fused exteins together through an amide bond (referred to as *cis*-splicing) [24]. SIs are a subgroup of inteins expressed from two genes as separate polypeptides, called N-intein and C-intein. The N- and C-inteins interact non-covalently and fold into a complex that cleaves themselves and ligates their fused exteins together into one protein (referred to as *trans*-splicing) [24]. The ligated extein product only contains a few residues from the SI (as few as six), which often do not affect the properties of the ligated PBMs.

Soon after their discovery, SIs were used to ligate individually folded protein domains into larger protein complexes [25]. Its value in synthesizing high-MW, highly repetitive PBMs was explored in recent years [3,26]. Dragline spider silk proteins such as MaSp1 contain more than one hundred repeats and have MWs of beyond 300 kDa. Previous microbial expression of MaSp1 using repetitive recombinant DNA can only synthesize silk proteins of 96 repeats, whose fibers were weak, only reaching 50% of the ultimate strength of natural spider silk fiber [27]. Using SI ligation, two separately expressed 96 repeats were ligated together, forming a silk protein that contained 192 repeats with a MW of 556 kDa [3]. Fibers spun from this high-MW silk protein displayed an ultimate strength of 1.03 ± 0.11 GPa and a toughness of 114 ± 51 MJ/m^3^, comparable with natural dragline spider silk from *Trichonephila clavipes*. This work has demonstrated that, for the first time, microbially produced recombinant silk protein fibers can fully replicate the mechanical performance of natural silk fibers [3].

Besides making highly repetitive silk proteins, SI-mediated ligation was also used to biosynthesize high-MW mussel foot proteins (Mfps) [2,28]. Mfps from *Mytilus galloprovincialis* display strong underwater adhesion for a wide range of surfaces, a feat that is difficult to achieve with most synthetic adhesives. Material studies have suggested that polymer adhesivity positively correlates with polymer chain length, as a longer adhesive polymers promote more extensive interactions between polymer and surface and between polymer chains. Thus, proteins containing multiple repeats of the Mfp sequence are expected to be more adhesive than natural Mfps. Unfortunately, microbial expression of Mfp trimer failed due to extremely low protein yield [2]. To solve this problem, SI was used to ligate a Mfp dimer with another Mfp (Figure 3a). The ligated Mfp trimer displayed strong underwater adhesion, with its force of adhesion 5.7-fold higher than that of recombinant Mfp monomer [2]. These oligomeric Mfp proteins were later used to form composite films by mixing with graphene oxide. Composite films made from higher MW Mfp oligomers displayed higher ultimate tensile strength and toughness than those made from Mfp monomer [28]. 

#### 4.1.2. Split Intein (SI)-Mediated Protein Polymerization

Beside one-step ligation, SI can also be used to perform multiple ligation steps in microbial cells, leading to the synthesis of protein polymers with extremely high MWs. To perform SI-mediated ligation multiple times, Bowen et al. developed a strategy called seeded chain-growth polymerization (SCP) that mimics chain-growth polymerization in synthetic polymer chemistry and works in living microbial cells (Figure 3b) [29]. In SCP, a silk protein monomer was fused with C-intein at its N-terminus and a N-intein at its C-terminus. The resulting “bifunctional” protein monomers (called Int^C^-silk-Int^N^) can ligate with each other at both chain ends to form polymers if their ligation reaction can be controlled to avoid self-cyclization. To do so, a seed protein that only contains the N-intein at its C-terminus (called Seed-Int^N^) was first expressed. The bifunctional monomer Int^C^-silk-Int^N^ was then expressed at a rate slower than the ligation rate, allowing the freshly synthesized C-intein of Int^C^-silk-Int^N^ to only react with Int^N^ of Seed-Int^N^ or a growing polymer chain of Seed-(silk)_n_-Int^N^, thus effectively preventing protein cyclization. As a result, high-MW, highly repetitive linear silk products with MWs up to 326 kDa were successfully produced in engineered *E. coli* (Figure 3c), indicating the linear elongation reaction can be performed for 15 rounds [29].

Additionally, SI-mediated protein polymerization was also used to biosynthesize animal muscle proteins with extremely high MWs (Figure 3c) [8]. In this work, a fragment of the rabbit soleus titin Ig67–70 was genetically fused with SI in the form of Int^C^-Ig67–70-Int^N^. Unlike flexible silk proteins, the folded Ig67–70 fragment is structurally rigid and its N- and C-termini are separated, 16.4 nm apart at opposite ends of the protein, thus minimizing the risk of intramolecular ligation to form cyclized proteins. Once expressed in *E. coli*, the monomeric Int^C^-Ig67–70-Int^N^ underwent multiple rounds of intracellular SI-catalyzed ligation, producing ultrahigh-MW titin polymers with an average size of 2.4 MDa [8]. The ultrahigh-MW titin polymers were then spun to fibers. These fibers displayed a high tensile strength of 378 ± 41 MPa and a high toughness of 130 ± 15 MJ/m^3^, 1.7- and 6.7-fold higher, respectively, than those of the monomeric Ig67–70 fibers.

#### 4.1.3. Sortase-Mediated Protein Ligation

Sortase is a transpeptidase that can be found in *Staphylococcus aureus* and other Gram-positive bacteria. Sortase recognizes a flexible peptide sequence (LPXTG) within a target protein and cleaves the peptide between the threonine and glycine residues in the presence of Ca^2+^ ions. The free carboxyl group of threonine then reacts with the amino group of an N-terminal glycine from another protein, thereby mediating the ligation of the two proteins by forming an amide bond (known as sortagging) (Figure 3d) [30,31]. The ligation efficiency of wild-type sortases is very low, often requiring equimolar proportions of the substrate and enzyme to avoid side reactions [32]. Additionally, the ligated product also contains the LPXTG sequence, which can be cleaved again by sortase to ligate it back to the cleaved fragment.

To use sortase-mediated ligation for biotechnology applications, wild-type sortase has to be engineered to optimize its performance. First, several efforts were made to increase the activity of sortase and to decrease its dependence on Ca^2+^ ions. These efforts include mutation of glycine residues in the Ca^2+^ binding site of the enzyme [33] and immobilization of sortases on a solid support to decrease the amount of enzyme required [34]. The reversible reaction can also be shifted towards the forward reaction by genetically fusing the substrate and sortase [35] and deactivating the recognition motif in the ligated product through secondary structure formation [36].

Sortases have been widely used to functionalize recombinant proteins and cyclize target proteins [37]. So far, sortase has not yet been used to ligate low-MW material proteins into high-MW proteins but it presents some engineering opportunities. For example, orthogonal sortases that recognize different peptide motifs as substrates were engineered [38]. Furthermore, sortase-mediated ligation of multiple protein monomers has been achieved using ‘ligation site switching’ which inactivates the recognition motif in products to eliminate reversible reactions [39]. These strategies may enable multiple ligations of PBMs, forming high-MW protein oligomers with a defined sequence order.

### 4.2. Crosslinking of Proteins Using Side-Chain Chemistry

Apart from enzymatic ligation that links the backbones of two separate proteins, proteins can also be conjugated together or crosslinked on their sidechains, either by enzymatic reactions between specific peptide motifs or chemical reactions on unique sidechain functional groups.

#### 4.2.1. Catcher/Tag Reactions

Catcher/Tag protein pairs catalyze spontaneous two-component conjugation reactions that covalently link each protein pair together. So far, three orthogonal Catcher/Tag pairs were developed, including SpyCatcher–SpyTag, SnoopCatcher–SnoopTag, and DogCacther–DogTag [40]. Upon Catcher and Tag recognition, the Catcher/Tag pair catalyzes the formation of an isopeptide bond between the amino group of a lysine sidechain in the Catcher and the carboxylate group of an aspartate or asparagine in the Tag (Figure 4a,b). Whereas Tag proteins are usually short peptides of approximately 10 residues, the size of Catcher proteins are typically 10-fold greater than that of Tags. If a Catcher/Tag pair are fused to two target proteins, the conjugation reaction can covalently link the two target proteins together, leaving the Catcher/Tag protein in between.

Catcher/Tag reactions have rapid reaction kinetics (often diffusion limited) [42], high yield, and high robustness under various conditions. However, Catcher/Tag-conjugated products unavoidably contain the large Catcher/Tag domain (SpyCatcher–SpyTag 10.5 kDa, SnoopCatcher–SnoopTag 14.0 kDa), which may affect the properties of many protein products.

Catcher/Tag chemistry has been widely used to link proteins of interest together [43,44,45,46]. The conjugation reaction was proven effective when performed either in test tubes or in living cells [47]. Catcher/Tag has also been used to create protein oligomers and multiple-protein conjugates [41,48]. When fused to elastin-like proteins (ELPs), oligomers containing up to five repeats were successfully produced (Figure 4c) together with lower MW oligomers and some circularized byproducts [41].

#### 4.2.2. Other Enzyme-Mediated Protein Conjugation Reactions

Other enzymes exist that recognize specific peptide motifs and form an amide bond between the amino acid side chains of two proteins. These enzymes have been employed for protein conjugation and have potential for the synthesis of HMW protein polymers. Here, we briefly discuss some of the most commonly used enzyme-mediated conjugation reactions.

*Formylglycine-generating enzyme (FGE)-mediated crosslinking.* FGE recognizes the pentapeptide CXPXR motif and converts the cysteine into formylglycine. The aldehyde group of formylglycine can then react reversibly with amines to form imines that can be irreversibly converted to a C-N single bond by strong reducing reagents (reductive imination) [49]. Furthermore, aldehyde can react with aminooxy or orthogonal reactive groups such as azide and alkyne groups to promote protein crosslinking [50].

*Lipoic acid protein ligase (LAPL)-mediated conjugation.* LAPL recognizes a 13-residue LAP sequence (GFEIDKVWYDLDA) and forms an amide bond between the amino group of lysine in the LAP sequence with a carboxylic acid group from glutamic acid or aspartic acid of another protein (Figure 5a) [51]. This reaction uses the inverse electron demand Diels–Alder (IEDDA) mechanism and proceeds with an approximate second-order rate constant of 50 M^−1^ s^−1^ at 37 °C in PBS.

*Small ubiquitin-like modifier (SUMO)-enzyme-mediated conjugation.* The SUMO enzyme recognizes the IKXE motif and forms an amide bond between the lysine of the IKXE motif and a C-terminal thioester from another protein [52]. However, this method is only suitable for the ligation of ubiquitin-like proteins. 

#### 4.2.3. Cysteine-Based Protein Crosslinking

The cysteine sidechain is one of the most commonly used targets for protein–protein crosslinking due to its unique reactivity from the sulfhydryl group and relatively low frequency of occurrence in most proteins (Figure 5b).

Cysteine-containing proteins can be crosslinked by linkers containing two or multiple maleimide groups [53]. Cysteine–maleimide conjugation is usually carried out at a neutral pH and room temperature over several hours. Cysteine sidechains can also be crosslinked by bifunctional linkers containing bromoacetamide or iodoacetamide groups [54]. Reactions between cysteine and haloacetamide require a basic pH to deprotonate the sulfhydryl group. Furthermore, cysteine can be converted to alkyne or azide by bifunctional chemicals, followed by crosslinking using click chemistry to form covalently linked protein complexes [50]. Click-chemistry-based crosslinking offers higher reaction kinetics and conjugation yield compared with other cysteine-based crosslinking reactions.

Effective protein crosslinking requires a bifunctional crosslinker to react with different protein molecules while preventing intramolecular crosslinking. This can be achieved when two cysteine residues are exposed at two different sides of a rigid protein molecule [55]. Furthermore, selecting a crosslinker with proper length and flexibility is also important. Whereas a linker that is too short cannot react with two different protein molecules, a linker that is too long may favor intramolecular reactions.

#### 4.2.4. Lysine Side Chain Modification

The lysine side chain is also vastly employed due to its high reactivity to various functional groups [56]. The amine group of lysine is often reacted with the carboxylate group of glutamate or aspartate to form a peptide bond. This reaction is rapid if catalyzed by 1-ethyl-3-(3-dimethylaminopropyl)-carbodiimide (EDC) and N-hydroxysuccinimide (NHS) (Figure 5c) [57]. The amine side chain of lysine can also be targeted by hetero-bifunctional crosslinkers such as N-hydroxysulfosuccinimide/aryl sulfonyl fluoride (NHSF) or N-hydroxysuccinimide/ortho-quinone methide (NHQM), which reacts with the nucleophilic Ser, His, Thr, or Tyr side chains in a neighboring monomer [58,59].

#### 4.2.5. Tyrosine Side Chain Oxidation

The phenol side chain of tyrosine can be oxidized to quinone by enzyme tyrosinases. Quinone can then react with multiple nucleophiles (such as amino or thiol groups) or with themselves through radical-based reactions to form conjugates [60]. These reactions can be used to crosslink proteins containing surface-exposed tyrosine residues (e.g., a C-terminal GGGGY motif) upon oxidization with enzymes such as laccases or horseradish peroxidases (Figure 5d) [61]. Furthermore, proteins with the C-terminal Tub sequence (VDSVEGEGEEEGEE) can be modified by the tubulin tyrosine ligase (TTL), which adds a tyrosine or azide/alkyne-functionalized tyrosine to the C-terminal end of the Tub sequence. The modified protein can then be crosslinked via click chemistry [62].

### 4.3. Comparing Different Protein Ligation and Conjugation Approaches to Synthesize Repetitive Protein Oligomers and Polymers

The above-discussed protein ligation and conjugation chemistry offers multiple choices to form covalently linked, higher-order protein complexes, either as liner repetitive protein oligomers/polymers or as crosslinked protein complexes. Each of methods has different features which are summarized in Table 2 and Table 3. Synthesis of linear protein oligomers/polymers requires a protein monomer to react with other monomers on both termini while preventing intramolecular reactions that lead to protein cyclization. So far, only SIs and Spy Catcher/Tag were used to synthesize linear, high-MW, repetitive protein polymers [2,3,8,29,41,63]. Comparing these two methods, SI is advantageous as its ligation only introduces 6–10 amino acid residues to the reaction sites, whereas Catcher/Tag reactions leave 10–14 kDa protein sequences in each conjugation site. Sortase-catalyzed ligation only requires short peptide sequences to be present in the product; however, the ligation efficiency of sortase-catalyzed reactions is currently low for protein polymerization.

For protein crosslinking strategies, most methods cannot offer sufficient control over the structure and MW of the crosslinked protein products. These methods are more useful for the synthesis of specific protein–protein dimers, cyclized proteins, and densely crosslinked protein hydrogels. Future engineering efforts are needed to tune these ligations reactions as efficient tools to form controlled protein polymers.

## 5. Conclusions

As demands for renewable, biodegradable, and mechanically advantageous materials increase, there is an urgent need to develop cost-effective approaches for the bioproduction of high-MW, highly repetitive PBMs in sufficient scale for material innovation.

Over the past few years, multiple strategies have been developed for the synthesis of repetitive DNAs and the polymerization of mRNAs and proteins. Among these strategies, protein polymerization is particularly promising as it avoids the issues related to repetitive DNAs and mRNAs. So far, published PBMs of the highest MW are in vitro-ligated recombinant spider silk (556 kDa, containing 192 repeats) [3] and in vivo-polymerized titin repeats (mass average MW of 2.4 MDa, containing ~50 repeats) [47]; both used SI chemistry, thus demonstrating power of this technique in producing highly repetitive PBMs. These ultrahigh-MW proteins have resulted in fibers with extraordinarily high ultimate tensile strength (up to 1.03 ± 0.11 GPa), high toughness (up to 130 ± 15 MJ/m^3^), high damping energy (53.3 ± 2.6 MJ/m^3^ at 30% strain), robust cyclic behavior, and other attractive mechanical properties, enabling a wide range of future applications. Protein polymerization in living microbial cells is particularly useful as it avoids performing ligation reactions using purified proteins. In vivo protein polymerization can be further engineered to improve its efficiency, polymer yields, and control over polymer MW distribution. Such goals are possible with support from modern synthetic biology. Synthetic biology has enabled the controlled expression of metabolic pathways containing more than a dozen enzymes [64,65], leading to the microbial production of complex chemicals, including unnatural and naturally rare compounds [66,67]. Compared with such engineering efforts, protein polymerization only requires the expression of a few genes, thus can be more easily controlled for the resultant protein. Additionally, synthetic biology has created numerous tools to regulate microbial gene expression and metabolism [68,69,70], promoting titers, yields, and productivities of microbially synthesized products, which will be important to facilitate the scalable production of PBMs for material applications.

In conclusion, many innovative strategies and tools have been developed for the biosynthesis of high-MW, highly repetitive PBMs. Several of these microbially produced PBMs have displayed mechanical properties similar to or greater than the best performing natural PBMs. Although these methods still have their own limitations and scalable production of truly high MW PBMs in high yields is still a challenge, we believe that future developments in synthetic biology will make biosynthesized PBMs a popular type of material for a wide range of applications.

## Figures and Tables

**Figure 1 ijms-24-06416-f001:**
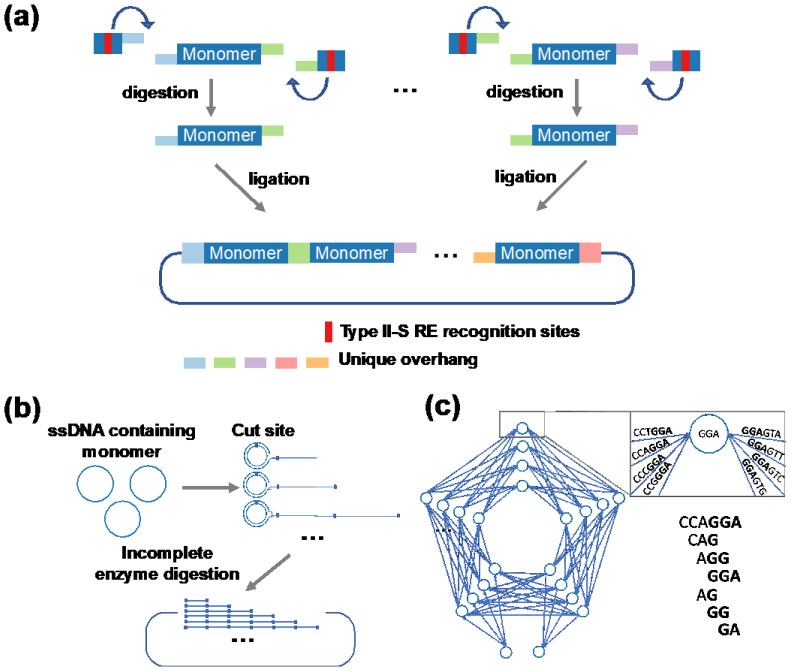
Methods for assembling repetitive DNA to encode repetitive proteins. (**a**) Golden Gate DNA assembly. (**b**) Rolling circle amplification. (**c**) Combinatorial codon scrambling for synthesizing repetitive DNAs [13,18,19].

**Figure 2 ijms-24-06416-f002:**
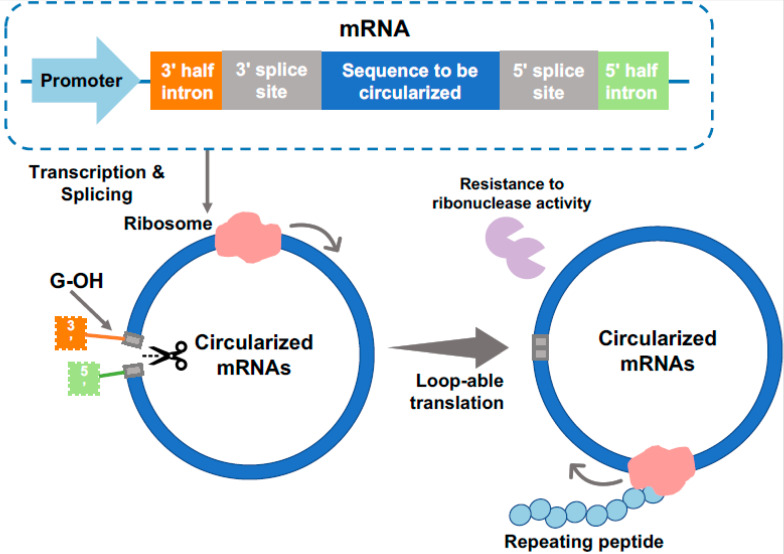
Synthesis of circular mRNA and its translation into repetitive proteins. Circular mRNA can be prepared using *td* intron. *Td* intron is a self-splicing intron from the thymidylate synthase (*td*) gene of the T4 bacteriophage. *Td* intron was divided and relocated, which placed the 3′ half intron and 3′ splice site upstream and a 5′ half intron and 5′ splice site downstream of a targeted protein sequence. The splicing reaction is catalyzed by exogenous guanosine and forms a back-splice junction. During circularization, the 5′ caps and 3′ tails are removed, which prevents the binding of ribonuclease and initiates the loopable translation [20,21].

**Figure 3 ijms-24-06416-f003:**
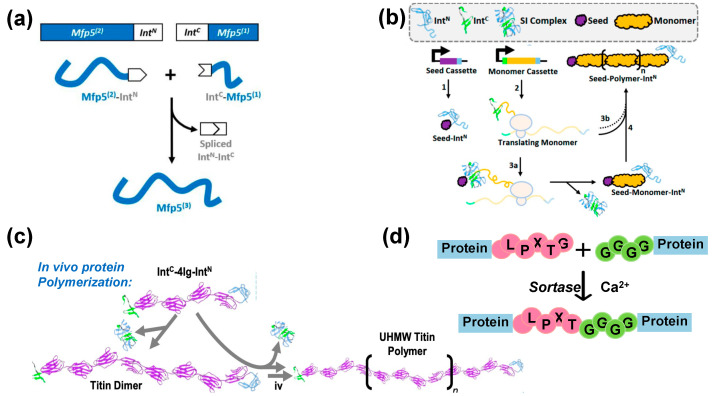
Different strategies for ligation of protein backbones via a peptide bond. (**a**) SI-mediated one-step protein–protein ligation of the mussel foot protein Mfp5 [2]. Copyright 2018, American Chemical Society. (**b**) SI-mediated chain growth protein polymerization [29]. Copyright 2019, American Chemical Society. (**c**) SI-mediated protein polymerization of titin Ig 67–70 [8]. Copyright 2021, Nature Publishing Group. (**d**) Sortase-mediated protein ligation reaction.

**Figure 4 ijms-24-06416-f004:**
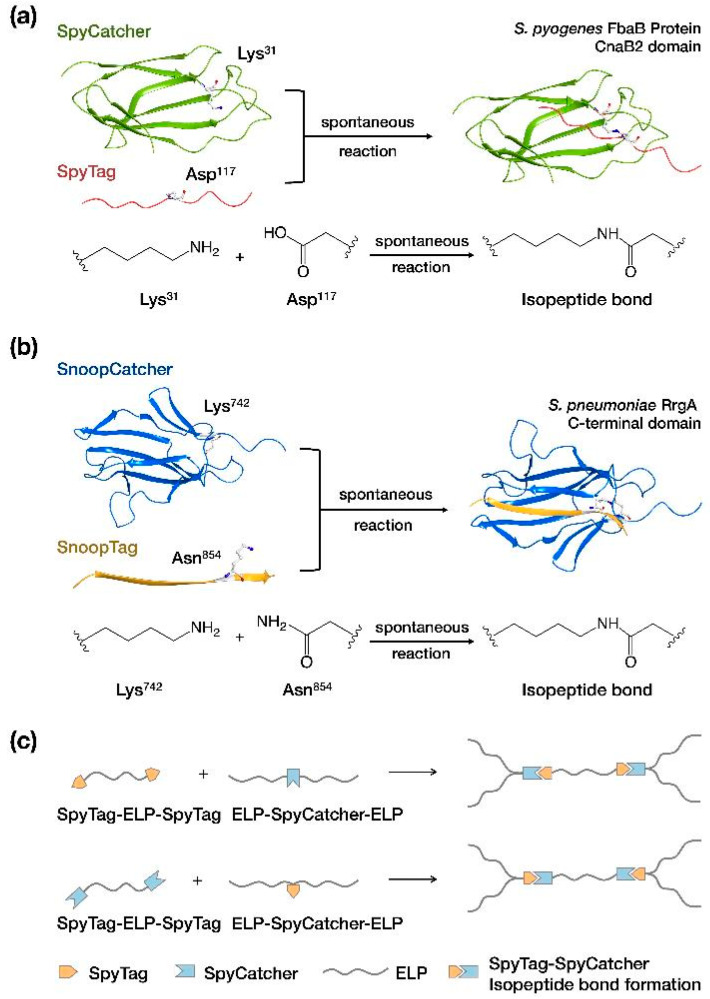
The spontaneous isopeptide bond formation by different Catcher/Tag reactions. (**a**) SpyTag–SpyCatcher (PDB entry: 4MLI) and (**b**) SnoopTag–SnoopCatcher (PDB entry: 2WW8). (**c**) An illustration of applying the SpyCatcher–SpyTag reaction into HMW highly repetitive protein creation [41].

**Figure 5 ijms-24-06416-f005:**
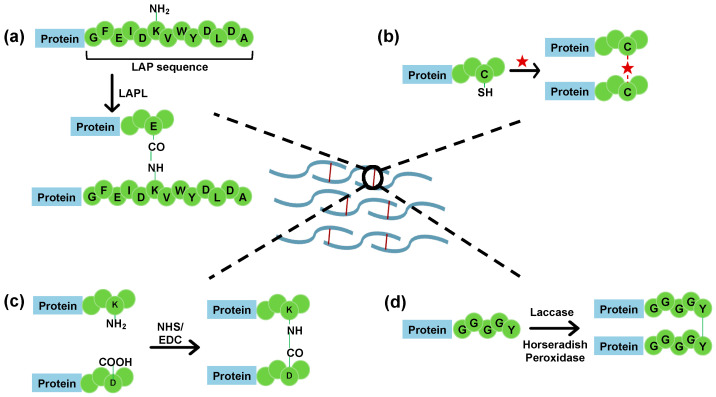
Various protein crosslinking strategies using side-chain chemistry. (**a**) Lipoic acid protein ligase mediated crosslinking that targets the LAP sequence for formation of an intermolecular amide bond. (**b**) Cysteine-specific conjugation using homobifunctional linkers such as maleimide (marked with red star in figure). (**c**) Lysine-specific crosslinking catalyzed by 1-ethyl-3-(3-dimethylaminopropyl)-carbodiimide (EDC) and N-hydroxysuccinimide (NHS). (**d**) Tyrosine-specific crosslinking using enzymes such as laccase and horseradish peroxidase for the oxidation of phenol side chains.

**Table 1 ijms-24-06416-t001:** Comparison between different methods for assembling repetitive DNAs. “+” and “++” symbols indicate the respective levels of time commitments.

	Golden Gate Assembly	Rolling Circle Amplification	Codon Scrambling	Circular mRNA
Unwanted DNA between monomers	No	Yes	No	Yes
Precise control of repeat numbers	Yes	No	Yes	No
Time consuming	+	++	+	+

**Table 2 ijms-24-06416-t002:** Comparison between different methods for ligation of the protein backbones via peptide bonds. “− “, “+” and “++” symbols indicate the respective levels of each criterion.

	Split Intein (SI) -Mediated Protein Ligation	Split Intein (SI)-Mediated Protein Polymerization	Sortase-Mediated Protein Ligation
Ligation rate	++	++	+
Ligation efficiency	++	+	−
High MW protein yield	+	++	−

**Table 3 ijms-24-06416-t003:** Comparison between different methods for crosslinking of proteins using side-chain chemistry. “− “, “+” and “++” symbols indicate the respective levels of each criterion.

	Catcher/Tag Reactions	Cysteine-Based Protein Crosslinking	Lysine Side Chain Modification	Tyrosine Side Chain Oxidation
Reaction rate	++	+	+	+
Conjugation yield	+	++	++	+
Have large conjugation domain left	Yes	No	No	No

## Data Availability

No new data were created or analyzed in this study. Data sharing is not applicable to this article.

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
