# Peer review of "Microbial Synthesis of High-Molecular-Weight, Highly Repetitive Protein Polymers"

_ijms, 2023, doi:10.3390/ijms24076416_

Round 1

Reviewer 1 Report

In this manuscript, Jeon et al review the strengths and weakness of different synthetic biology methods to produce highly repetitive protein polymers (protein-based materials) to replace petroleum-dervied materials. The authors have further broken down the challenges into the construction of genes, expression and post-translational ligation.   Overall there are strong points
  1. Very comprehensive and cogent discussion of various strategies
  2. Highlights strengths and weaknesses inherent to each approach
  Steps to strengthen the manuscript
  1. Inclusion of Tables like Table1 for other sections Protein Ligation (yield, kinetics etc) would be helpful for users to get a comparative picture of different approaches quickly
  Minor points
  1. Typo in Abstract - PBMs are renewable, biodegradation (biodegradable)
  Overall I think this paper provides a useful review of synthetic biology approaches in developing PBMs

Author Response

Point-by-point Response to Reviewer’s Comments

Reviewer1:

We appreciate your effort on reviewing our manuscript and give detailed comments, which has helped us improve our manuscript.

  1. Inclusion of Tables like Table1 for other sections Protein Ligation (yield, kinetics etc.) would be helpful for users to get a comparative picture of different approaches quickly.

We agree that inclusion of tables could help the readers understanding. We have included new tables in the revised manuscript.

  1. Typo in Abstract - PBMs are renewable, biodegradation (biodegradable)

Thank you for the careful review of the manuscript. The typo in abstract has been corrected.

Reviewer 2 Report

The review paper tackles the important challenge of producing protein polymers in microbial hosts. Great attention has been given to spider silk fibers in the introduction as one relevant example. Afterward, the manuscript addresses relevant methods and includes a critical review of such. The language is clear and concise, being easier to read and follow throughout the text.

There is a focus on methods, and maybe this should be emphasised in the title and abstract.

Observation lines 85 -86 - have not been deleted from the template.

For a review, it could be longer. But in any case, it was very clear, well-written and very informative.

Author Response

Point-by-point Response to Reviewer’s Comments

Reviewer2:

We appreciate your effort on reviewing our manuscript and give detailed comments, which has helped us improve our manuscript.

1.There is a focus on methods, and maybe this should be emphasized in the title and abstract.

Thanks for this comment. We have revised our abstract to emphasize the focus of methods.

  1. Observation lines 85 -86 - have not been deleted from the template.

We thank you for the careful reviews of the manuscript. The observation lines were deleted, and other lines in manuscripts were also double checked.